# High-flow nasal cannula for reducing hypoxemic events in patients undergoing bronchoscopy: A systematic review and meta-analysis of randomized trials

Chien-Ling Su[1,2]☯, Ling-Ling Chiang[1]☯, Ka-Wai Tam[3,4,5]☯, Tzu-Tao Chen[1]*, Ming-Chi Hu[1]*

**1** Division of Pulmonary Medicine, Department of Internal Medicine, Shuang Ho Hospital, Taipei Medical University, New Taipei City, Taiwan, **2** Department of Physical Therapy, Shu-Zen Junior College of Medicine and Management, Kaohsiung City, Taiwan, **3** Division of General Surgery, Department of Surgery, Shuang Ho Hospital, Taipei Medical University, New Taipei City, Taiwan, **4** Division of General Surgery, Department of Surgery, School of Medicine, College of Medicine, Taipei Medical University, Taipei, Taiwan, **5** Cochrane Taiwan, Taipei Medical University, Taipei, Taiwan

☯ These authors contributed equally to this work.
* 09330@s.tmu.edu.tw (T-TC); 20549@s.tmu.edu.tw (M-CH)

**Data Availability Statement:** All relevant data are within the manuscript and its Supporting Information files.

## Abstract

### Background

Patients undergoing bronchoscopic procedures may develop hypoxemia and severe complications. High-flow nasal cannula (HFNC) may prevent hypoxemic events during bronchoscopy. We conducted a systematic review of randomized controlled trials (RCTs) to evaluate the effectiveness of HFNC in these patients.

### Methods

We conducted a search in PubMed, Embase, and the Cochrane Library for RCTs published before November 2021. Individual effect sizes were standardized, and a meta-analysis was performed to calculate the pooled effect size using random-effects models. The primary outcome was the incidence of hypoxemic events (oxygen saturation [$SpO_2$] < 90%) during bronchoscopy. Secondary outcomes included the incidence of interrupted bronchoscopy due to desaturation, lowest $SpO_2$ during bronchoscopy, partial pressure of oxygen ($PaO_2$), partial pressure of carbon dioxide ($PaCO_2$), end-tidal $CO_2$ ($EtCO_2$) at the end of bronchoscopy, and the incidence of intubation after the procedure.

### Results

Five trials involving 257 patients were reviewed. The incidence of hypoxemic events was lower in the HFNC group than in the conventional oxygen therapy group (risk ratio, 0.25; 95% confidence interval [CI], 0.14–0.42). The lowest $SpO_2$ during the procedure was significantly higher in the HFNC group than in the conventional oxygen therapy group (weighted mean difference [WMD], 7.12; 95% CI, 5.39–8.84). $PaO_2$ at the end of the procedure was

**Funding:** The authors have received no specific funding for this work.

**Competing interests:** The authors have no conflicts of interest or financial associations to disclose.

significantly higher in the HFNC group than in the conventional oxygen therapy group (WMD, 20.36; 95% CI, 0.30–40.42). The incidence of interrupted bronchoscopy due to desaturation, $PaCO_2$ and $EtCO_2$ at the end of the procedure, and the incidence of intubation after the procedure were not significantly different between groups.

## Conclusions

HFNC may reduce the incidence of hypoxemic events and improve oxygenation in patients undergoing bronchoscopy.

## Introduction

Hypoxemia is one of the complications in patients undergoing bronchoscopy. Sedation and occlusion of the bronchi during the procedure reduce the respiratory drive and lead to hypoventilation [1]. Patients with pulmonary complications after bronchoscopic procedures occasionally have a risk of hypoxemic events that require rescue airway interventions [2]. Complications of bronchoscopy, such as refractory hypoxemia and respiratory depression, can be debilitating without careful monitoring [2–4]. Therefore, oxygen supplementation during bronchoscopy is crucial for these patients.

In the past, conventional oxygen therapy was usually adopted for patients undergoing diagnostic or therapeutic bronchoscopy, but desaturation would occasionally occur due to impaired respiratory drive and hypoventilation [2–4]. For safer bronchoscopic procedures, high-flow nasal cannula (HFNC) may replace conventional oxygen supply due to the more consistent fraction of inspired oxygen [5, 6]. According to a previous study, in patients using conventional oxygen therapy, the complication rate can reach 35% after bronchoscopy [7]. Thus, conventional oxygen therapy may not be as useful as expected for maintaining oxygenation.

Currently, HFNC is used to prevent respiratory failure. HFNC enhances secretion clearance and reduces bronchoconstriction by using heated and humidified gas [8], decreases dead space [9], and restores functional residual capacity to improve ventilation/perfusion mismatch (V/Q mismatch) [10]. In the past, several feasibility studies and reviews have assessed whether HFNC can maintain oxygenation and avoid endotracheal intubation in patients undergoing bronchoscopy [7, 11, 12]. Since then, several randomized controlled trials (RCTs) have been reported [13–17], but no systematic review or meta-analysis has encompassed them all. Thus, the aim of this systematic review was to compare the efficacy of HFNC with conventional oxygen therapy in maintaining oxygenation ($SpO_2 \geq 90\%$) and avoiding endotracheal intubation in patients undergoing bronchoscopy.

## Materials and methods

### Inclusion criteria

RCTs that compared HFNC with conventional oxygen therapy in patients undergoing bronchoscopy were included in the analysis. We excluded trials in which (1) patients were <18 years old, (2) patients underwent bronchoscopy for intubation, (3) comparisons different from the one of interest were performed, or (4) duplicate patient cohorts were reported.

## Search strategy and study selection

Relevant trials published before November 2021 were identified from the PubMed, Embase, and Cochrane Library databases. The following Medical Subject Headings (MeSH) were used in the search: "HFNC", "high flow nasal cannula," "high flow oxygen," "high flow nasal oxygen," "bronchoscopy," and "bronchoscope." These were combined into the search strategy detailed in S1 Appendix. No language restrictions were imposed. The ClinicalTrials.gov registry was searched for ongoing trials. This systematic review was registered in the online PROSPERO International Prospective Register of Systematic Reviews of the National Institute for Health Research (registration number CRD42021254176). The completed PRISMA checklist was provided in S2 Appendix.

## Data extraction

Two authors independently extracted baseline and outcome data, study designs, study population characteristics, inclusion and exclusion criteria, HFNC settings, sedative or anesthetic agents, and post-treatment parameters from the studies retrieved by the database search. The reviewers' individually recorded decisions were compared, and disagreements concerning data extraction were resolved through discussion with a third reviewer.

## Methodological quality appraisal

Two reviewers independently assessed the methodological quality of each study using the risk-of-bias tool, version 2, as recommended by the Cochrane Collaboration [18]. For randomized trials, we assessed allocation, performance, attrition, measurement, reporting, and overall bias. Disagreements regarding the assessment of risk of bias were resolved through a comprehensive discussion.

## Outcomes

The primary outcome was the incidence of hypoxemic events during bronchoscopy (oxygen saturation [$SpO_2$] < 90%). Secondary outcomes included the incidence of interrupted bronchoscopy due to desaturation, lowest $SpO_2$ during bronchoscopy, partial pressure of oxygen ($PaO_2$), partial pressure of carbon dioxide ($PaCO_2$), end-tidal $CO_2$ ($EtCO_2$) at the end of bronchoscopy, and the incidence of intubation after the procedure.

## Grading evidence quality

Two reviewers independently assessed the evidence quality for each outcome using the Grading of Recommendations Assessment, Development, and Evaluation (GRADE) guidelines [19]. Evidence quality was classified as high, moderate, low, or very low based on the assessed risk of bias, inconsistency, indirectness, imprecision, and publication bias. Discrepancies were resolved by consensus.

## Statistical analyses

We analyzed the data using Review Manager version 5.4 (Cochrane Collaboration, Oxford, England) in accordance with the PRISMA guidelines [20]. Risk ratios (RRs) for binary outcomes and weighted mean differences (WMDs) for continuous outcomes with the corresponding 95% confidence intervals (CIs) were computed. Standard deviations were estimated from CI limits or standard errors. If the mean and variance were not reported in a trial, they were estimated from the median, interquartile range (IQR), and sample size if the skewness was acceptable [21]. All outcomes were analyzed using a random-effects model [22]. Cochran's

$Q$ and $I^2$ statistics were calculated to evaluate the statistical heterogeneity and inconsistency of treatment effects, respectively, across trials. Statistical significance was set at $p < .10$ for Cochran's $Q$ tests. Statistical heterogeneity across trials was assessed using the $I^2$ test, which quantifies the proportion of total outcome variability across trials [23]. Trial sequential analysis was performed to reduce type I errors, and sensitivity analysis was used to manage heterogeneity [18, 24, 25].

## Results

### Trial characteristics

Fig 1 illustrates the study selection process. The initial search yielded 371 citations. After removal of duplicates, 308 studies were left, of which 270 did not compare HFNC with conventional oxygen therapy in patients undergoing bronchoscopy. Among these reports, nine registrations that were not retrieved were excluded. After the remaining 29 studies were reviewed, 24 were excluded. Of these, 18 were not RCTs, two included patients aged <18 years, two investigated bronchoscopy for intubation, and two conducted different comparisons. Hence, five trials were eligible for this study [10–14]. The search strategy and list of 23 major exclusions are reported in the S1 Appendix.

Five trials on patients undergoing bronchoscopy were published between 2012 and 2021 [13–17]. They had sample sizes ranging from 36 to 76, with a total sample size of 257. All the trials enrolled adult patients with indications for diagnostic or therapeutic interventions. Three trials measured baseline oxygenation by examining $SpO_2$ [13–15], one by investigating $PaO_2$ [16], and one by examining $PaO_2/FiO_2$ [17]. Ben-Menachem et al. performed local anesthesia with nebulized 2% lidocaine and sedation with midazolam, propofol, and alfentanil [13]. Douglas et al. conducted sedation with midazolam, opioids, or propofol [14]. Irfan et al. conducted moderate sedation with midazolam and alfentanil [15]. Longhini et al. administered an anesthetic spray containing 10% lidocaine [16]. Lucangelo et al. performed local anesthesia nebulized lidocaine 2% through the mouth and nostrils [17]. Notably, Ben-Menachem et al. investigated post-lung transplant patients [13]. All the included trials categorized patients into two groups: those undergoing HFNC and those undergoing conventional oxygen therapy. Regarding intervention timing, HFNC was conducted during bronchoscopy in all trials. The baseline characteristics of the patients in each trial are summarized in Table 1.

The methodological quality of the included studies is summarized in Table 2. Regarding allocation bias, one trial presented an imbalance in the comorbidity of pulmonary carcinoma and $SpO_2$ following preoxygenation at baseline [14], one trial showed an imbalance for the diagnosis of metastatic cancer at baseline [15], and one trial did not provide information on age [16]. All trials had acceptable management of performance, attrition, measurement, and reporting biases. Overall, two trials were rated as having a low risk of bias [13, 17], and three trials had some concerns regarding bias [14–16].

### Incidence of hypoxemic events during bronchoscopy and of interrupted bronchoscopy due to desaturation

A total of four trials measured the incidence of hypoxemic events after bronchoscopy [13–16]. All trials defined hypoxemic events as $SpO_2 < 90\%$. The incidence of hypoxemic events was significantly lower among patients who underwent HFNC than among those who underwent conventional oxygen therapy (RR, 0.25; 95% CI, 0.14–0.42; Fig 2). Furthermore, the results of trial sequential analysis showed that the Z-curve crossed the O'Brien-Fleming boundaries after the fifth cumulative significance testing. These findings indicate that HFNC may reduce

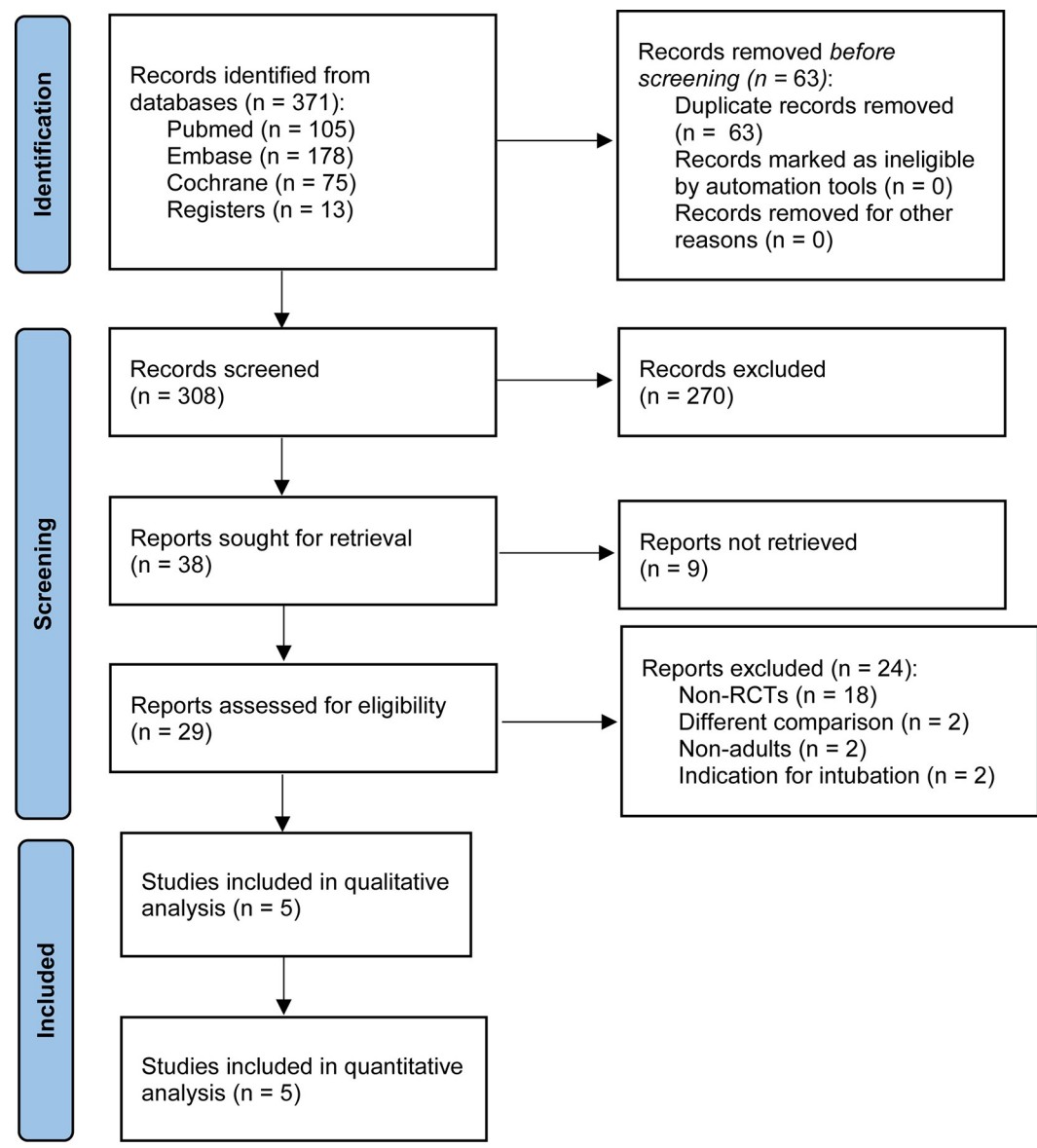

**Fig 1. Flowchart of study selection.** RCT, randomized controlled trial.

hypoxemic events ($SpO_2 < 90\%$) in patients undergoing bronchoscopy (S3 Appendix). Two trials measured the incidence of interrupted bronchoscopy due to desaturation [13, 14], which was lower among patients who underwent HFNC than among those who underwent conventional oxygen therapy (RR, 0.19; 95% CI, 0.02–1.86), although the result was not statistically significant (Fig 2).

## Lowest SpO$_2$ during bronchoscopy

Four trials measured the lowest $SpO_2$ during the procedure in patients undergoing bronchoscopy [13–16]. The lowest $SpO_2$ was significantly higher in patients on HFNC than in those on conventional oxygen therapy (WMD, 7.12; 95% CI, 5.39–8.84; Fig 3).

**Table 1. Characteristics of the selected randomized controlled trials.**

| Study | Inclusion criteria | No. of patients (% male) | Age, years, mean ± SD | Baseline oxygenation | Sedative or anesthetic agents; duration of bronchoscopy (min) | Interventions |
|---|---|---|---|---|---|---|
| Ben-Menachem [13] | Age ≥ 18 years; lung transplant recipients; undergoing TBLB; able to provide informed consent; English speaking | H: 37 (40.5) | H: 54.9 ± 11.7 | H: 98 (97–99)[†,a] | Local Topicalized anesthesia with nebulized 2% lidocaine and midazolam (1 to 3 mg) sedation with midazolam, propofol (321 mg in intervention group and 337 mg in control group) and alfentanil (586 mcg in intervention group and 691 mcg in control group) to keep ASA score II–III; H: 33 ± 10, C: 34 ± 8 | H: FiO$_2$: 100%, flow rate: 30–50 LPM through the nasal cannula |
| | | C: 39 (25.6) | C: 55.8 ± 11.9 | C: 98 (97–99)[†,a] | | C: Flow rate: 4–10 LPM through standard oxygen tubing |
| Douglas [14] | Age ≥ 18 years; able to provide informed consent; sedation planned; English speaking | H: 30 (63) | H: 62.8 ± 14.1 | H: 96 (95–99)[†,a] | Topical 2% lignocaine to patient's nasopharynx and oropharynx Sedation with midazolam, opioids or propofol to keep MOAA/S = 4; I: 24 (26–28)[†], C: 21 (17–32)[†] | H: FiO$_2$: 100%, flow rate: 30–50 LPM (up to 70 LPM if necessary) through the nasal cannula |
| | | C: 30 (63) | C: 63.4 ± 14.3 | C: 96 (94–98)[†,a] | | C: Flow rate: 10 LPM (up to 15 LPM if necessary) through the bite block |
| Irfan [15] | Age ≥ 18 years; SpO$_2$ ≥ 90%; able to breathe spontaneously throughout the procedure | H: 20 (60) | H: 61.9 ± 12 | H: 98.4 ± 2.7[a] | Local anesthesia sedation with midazolam (5.6 mg in intervention group and 5.5 mg in control group) and alfentanil (300 mg in intervention group and 287 mcg in control group) varied by assessing purposeful response to verbal and/or tactile stimuli while preserving spontaneous respiratory efforts; NI | H: FiO$_2$: 36%, flow rate: 30 LPM through the nasal cannula |
| | | C: 20 (60) | C: 64.5 ± 14 | C: 96.9 ± 1.9[a] | | C: Nasal prong to maintain SpO$_2$ ≥ 94% |
| Longhini [16] | Age ≥ 18 years; outpatients undergoing flexible bronchoscopy for bronchoalveolar lavage | H: 18 (83) | H: 61.9 ± NI | H: 10.8 (8.7–12.0)[†,b] | Topical Aanesthetic spray containing 10% lidocaine over tongue and nasopharynx; gargles with 10 mL of 2% lidocaine hydrochloride solution guaranteed further anesthesia of the oropharynx; H: 11 min 30 s ± NI, C: 12 min 50 s ± NI | H: FiO$_2$ set to reach SpO$_2$ ≥ 95%, flow rate: 60 LPM through the nasal cannula |
| | | C: 18 (67) | C: 64.5 ± NI | C: 11.1 (10.5–12.1)[†,b] | | C: Nasal cannula to keep SpO$_2$ ≥ 94% |
| Lucangelo [17] | Age ≥ 18 years; BMI ranging from 21 to 30 | H60: 15 (47) | H60: 64 (63–70)[†] | H60: 350.9 (304.3–363.8)[†,c] | Anesthesia by nNebulized 2% lidocaine 2% through the mouth and nostrils to guarantee fully developed local anesthesia; 4mg midazolam in each group delivered as demanded by each patient, reaching a maximum dose of 0.1 mg/kg BW; H60: 15 (9–21)[†], H40: 15 (12–16)[†], C: 14 (10–16)[†] | H60: FiO$_2$: 50%, flow rate: 60 LPM through the nasal cannula |
| | | H40: 15 (53) | H40: 70 (61–76)[†] | H40: 342.8 (295.7–371.9)[†,c] | | H40: FiO$_2$: 50%, flow rate: 40 LPM through the nasal cannula |
| | | C: 15 (60) | C: 68 (62–78)[†] | C: 322.4 (295.6–374.3)[†,c] | | C: FiO$_2$: 50%, flow rate: 40 LPM through the venturi mask |

†Data reported as median (IQR); [a]Data reported as SpO$_2$; [b]Data reported as PaO$_2$ in kPa; [c]Data reported as PaO$_2$/FiO$_2$ ratio.

Abbreviations: BMI, body mass index; C, conventional oxygen therapy; H, high-flow nasal cannula; H60, high-flow nasal cannula with flow of 60 LPM; H40, high-flow nasal cannula with flow of 40 LPM; LPM, liters per minute; MOAA/S, Modified Observer's Assessment of Alertness/Sedation Scale; NI, no information; TBLB, transbronchial lung biopsy.

## PaO$_2$ at the end of bronchoscopy

Two trials measured PaO$_2$ at the end of the procedure in patients undergoing bronchoscopy [16, 17]. We compared the HFNC group with a flow rate of 60 liters per minute (LPM) with the conventional oxygen therapy group in Lucangelo's study [17]. The unit was converted from kPa to mmHg. PaO$_2$ was significantly higher in patients on HFNC than in those on conventional oxygen therapy (WMD, 20.36; 95% CI, 0.30–40.42; Fig 4).

**Table 2. Methodological quality assessment of the randomized trials.**

| Randomized controlled trials evaluated using the revised Cochrane Risk of Bias (RoB 2.0) tool | | | | | | |
|---|---|---|---|---|---|---|
| Study | Allocation bias | Performance bias | Attrition bias | Measurement bias | Reporting bias | Overall bias |
| Ben-Menachem [13] | Low risk | Low risk[d] | Low risk | Low risk | Low risk | Low risk |
| Douglas [14] | Some concerns[a] | Low risk | Low risk | Low risk | Low risk | Some concerns |
| Irfan [15] | Some concerns[b] | Low risk | Low risk | Low risk | Low risk[e] | Some concerns |
| Longhini [16] | Some concerns[c] | Low risk | Low risk | Low risk | Low risk | Some concerns |
| Lucangelo [17] | Low risk | Low risk | Low risk | Low risk | Low risk[e] | Low risk |

[a]The baseline of comorbidity of pulmonary carcinoma, procedural data, and the $SpO_2$ following pre-oxygenation were imbalanced.

[b]The baseline of diagnosis of metastatic cancer was imbalanced.

[c]The baseline of age was not available.

[d]One patient crossed over from the conventional oxygen group to the HFNC group in Ben-Menachem's trial, but they performed adequate analysis on an intend-to-treat (ITT) basis. Hence, we did not raise the risk of bias.

[e]Although these two trials did not provide protocol registration, they investigated the same outcomes as other trials registered in ClinicalTrials.gov. Obvious selective reporting was not detected. Hence, we did not raise the risk of bias.

## PaCO$_2$ and EtCO$_2$ at the end of the procedure

Two trials each measured $PaCO_2$ [16, 17] and $EtCO_2$ [14, 15] at the end of the procedure in patients undergoing bronchoscopy. We compared the HFNC group with a flow rate of 60 LPM with the conventional oxygen therapy group in Lucangelo's study [17]. The unit was converted from kPa to mmHg. $PaCO_2$ was higher among patients on conventional oxygen therapy than among those on HFNC (WMD, −0.02; 95% CI, −2.31–2.27). $EtCO_2$ was also higher in the conventional oxygen therapy group than in the HFNC group (WMD, −0.12; 95% CI, −4.19–3.94). However, these differences were not significant (Fig 5).

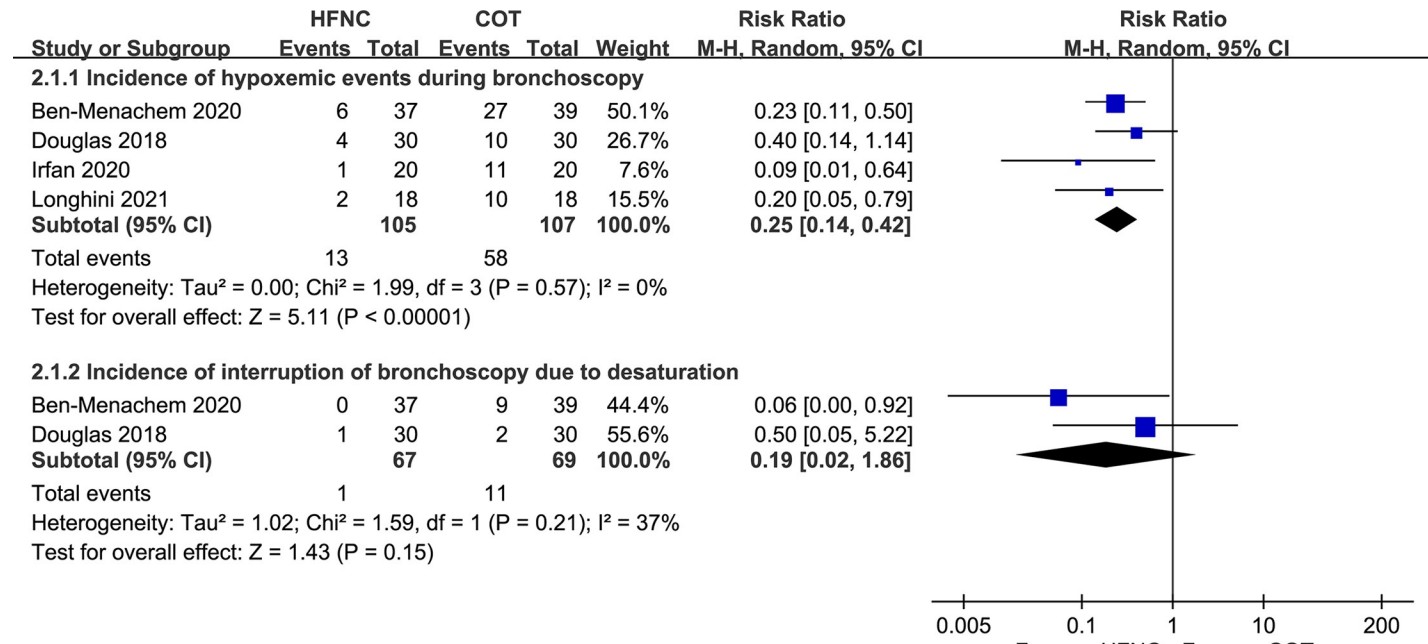

**Fig 2. Forest plot for comparison: High-flow nasal cannula versus conventional oxygen therapy.** Outcomes: incidence of hypoxemic events and incidence of interrupted bronchoscopy due to desaturation. HFNC, high-flow nasal cannula; COT, conventional oxygen therapy; CI, confidence interval; M-H, Mantel-Haenszel.

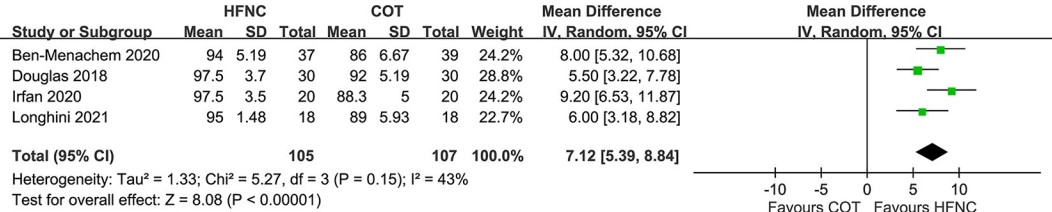

**Fig 3. Forest plot for comparison: High-flow nasal cannula versus conventional oxygen therapy.** Outcome: lowest $SpO_2$ during bronchoscopy. HFNC, high-flow nasal cannula; COT, conventional oxygen therapy; CI, confidence interval; SD, standard deviation.

### Incidence of intubation after the procedure

Two trials measured the incidence of intubation after bronchoscopy [14, 15]. No adverse events of endotracheal intubation were reported in either group. The incidence of intubation in both arms was 0.

### GRADE evidence quality

The GRADE evidence quality for each main outcome is shown in Table 3. In the risk-of-bias domain, all items were rated as serious because of the risk of bias in allocation. In the inconsistency and indirectness domains, all items were rated as having low risk because heterogeneity was acceptable among the trials, and all used head-to-head comparison. In the imprecision domain, the incidence of interrupted bronchoscopy due to desaturation and $PaO_2$, $PaCO_2$, and $EtCO_2$ at the end of the procedure were rated as serious due to imprecision attributable to an insufficient number of trials with a wide 95% confidence interval. No publication bias was observed. Thus, we obtained evidence of low quality for the incidence of interrupted bronchoscopy due to desaturation and $PaO_2$, $PaCO_2$, and $EtCO_2$ at the end of the procedure and of moderate quality for the incidence of hypoxemic events ($SpO_2 < 90\%$) and lowest $SpO_2$ during bronchoscopy.

## Discussion

Our findings indicate that in patients undergoing bronchoscopy, regardless of the sedative or anesthetic agents used, HFNC resulted in a lower incidence of hypoxemic events ($SpO_2 < 90\%$) when compared with conventional oxygen therapy, and it improved the values of the lowest $SpO_2$ during bronchoscopy and $PaO_2$ at the end of the procedure. The incidence of interrupted bronchoscopy, $PaCO_2$, $EtCO_2$, and the incidence of intubation did not differ significantly between groups. However, the evidence was limited by the low to moderate quality of these studies as assessed by the GRADE system.

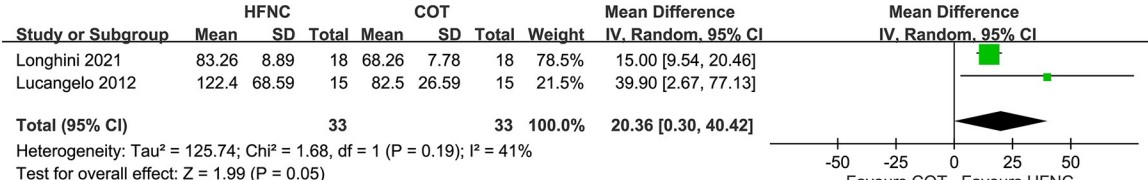

**Fig 4. Forest plot for comparison: High-flow nasal cannula versus conventional oxygen therapy.** Outcome: $PaO_2$ at the end of bronchoscopy. HFNC, high-flow nasal cannula; COT, conventional oxygen therapy; CI, confidence interval; SD, standard deviation.

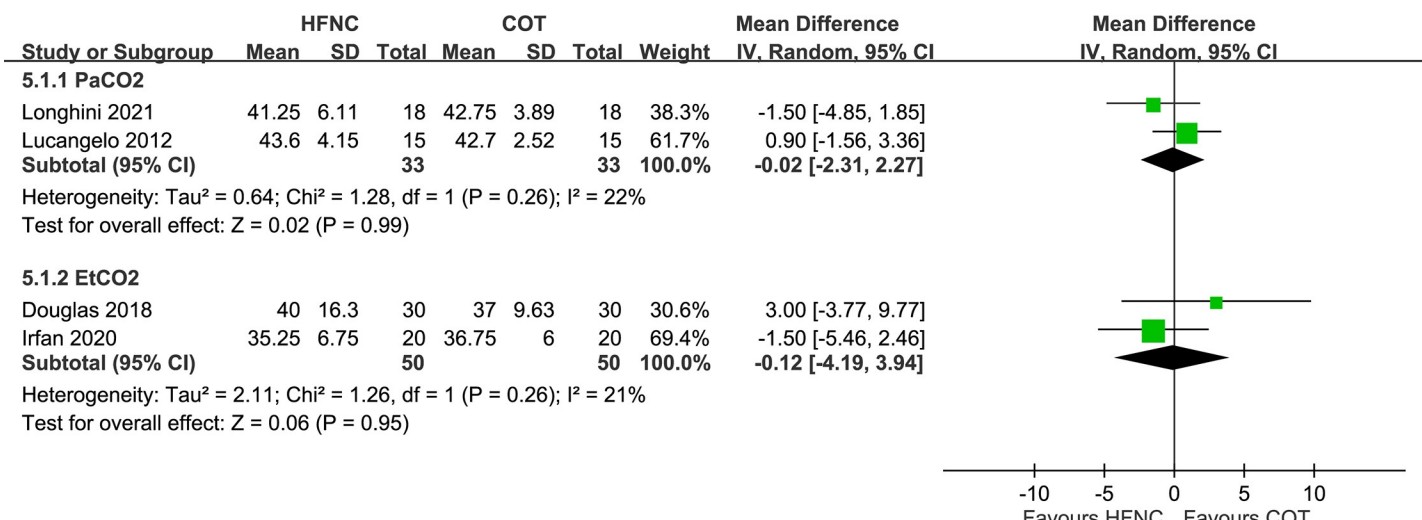

**Fig 5. Forest plot for comparison: High-flow nasal cannula versus conventional oxygen therapy.** Outcomes: $PaCO_2$ and $EtCO_2$ at the end of the procedure. HFNC, high-flow nasal cannula; COT, conventional oxygen therapy; CI, confidence interval; SD, standard deviation.

**Table 3. Summary of findings compiled using GRADE methodology.**

| Outcome | No. of studies | Study design | Risk of bias | Inconsistency | Indirectness | Imprecision | Publication bias | Effect size (95% CI) | Certainty | Importance |
|---|---|---|---|---|---|---|---|---|---|---|
| Incidence of hypoxemic events ($SpO_2$ < 90%) | 4 | RCT | Serious[a] | Not serious | Not serious | Not serious | Undetected | RR: 0.25 (0.14 −0.42) | ⊕⊕⊕○ Moderate | Critical |
| Incidence of interrupted bronchoscopy due to desaturation | 2 | RCT | Serious[a] | Not serious | Not serious | Serious[b] | Undetected | RR: 0.19 (0.02 −1.86) | ⊕⊕○○ Low | Important |
| Lowest $SpO_2$ during bronchoscopy | 4 | RCT | Serious[a] | Not serious | Not serious | Not serious | Undetected | WMD: 7.12 (5.39– 8.84) | ⊕⊕⊕○ Moderate | Important |
| $PaO_2$ at the end of bronchoscopy | 2 | RCT | Serious[a] | Not serious | Not serious | Serious[b] | Undetected | WMD: 20.36 (0.30– 40.42) | ⊕⊕○○ Low | Important |
| $PaCO_2$ at the end of bronchoscopy | 2 | RCT | Serious[a] | Not serious | Not serious | Serious[b] | Undetected | WMD: −0.02 (−2.31– 2.27) | ⊕⊕○○ Low | Important |
| $EtCO_2$ at the end of bronchoscopy | 2 | RCT | Serious[a] | Not serious | Not serious | Serious[b] | Undetected | WMD: −0.12 (−4.19– 3.94) | ⊕⊕○○ Low | Important |

Abbreviations: CI, confidence interval; GRADE, quality of evidence grade; NI, no information; RCT, randomized controlled trial; RR, risk ratio; WMD, weighted mean difference; ⊕⊕⊕○, moderate certainty; ⊕⊕○○, low certainty.

[a]Data reported as downgraded because of some concerns of bias.

[b]Data reported as downgraded because of wide CI or insufficient studies.

Patients undergoing bronchoscopy are more vulnerable to desaturation resulting from hypoventilation [2–4]. To overcome the patients' peak inspiratory demand flow and relieve their respiratory distress during bronchoscopy, a high-flow system can provide additional support [26, 27]. First, the heated humidification of inhaled gas may enhance bronchial hygiene and reduce bronchoconstriction [8]. Second, washout of the upper airway may reduce dead space [9]. Third, positive airway pressure may restore atelectatic lung regions [10]. Fourth, decreased entrainment of ambient air may increase oxygen supply [27]. Thus, HFNC may be a safe alternative to conventional oxygen therapy in patients undergoing bronchoscopy.

A reduced incidence of hazardous hypoxemic events ($SpO_2 < 90\%$) has been observed in both therapeutic and diagnostic bronchoscopy [13–16]. In our systematic review, Douglas et al. and Irfan et al. evaluated the efficacy of HFNC in patients undergoing endobronchial ultrasound procedure [14, 15], and Longhini et al. and Lucangelo et al. investigated the efficacy of HFNC in patients undergoing bronchoscopy for bronchoalveolar lavage (BAL) [16, 17]. The procedure duration varied among the trials and ranged from 11 min 30 s to 34 min. Regardless of the procedure performed, the lowest $SpO_2$ during bronchoscopy and the $PaO_2$ at the end of the procedure were higher in the HFNC group than in the conventional oxygen therapy group. Notably, with the exception of the trials by Douglas et al. and Lucangelo et al. [14, 17], all included trials showed that the lowest $SpO_2$ values were less than 90% in the control groups. Thus, the pooled results indicated that HFNC may improve patient safety and provide clinical benefits to patients undergoing bronchoscopy.

Both low and high pulmonary risk patients undergoing bronchoscopy may benefit from HFNC according to our inclusion criteria, because desaturation may occur at any level of forced expiratory volume 1 (FEV1), even without sedation [28]. Among our included trials, Ben-Menachem et al. investigated the same issues in patients after lung transplantation [13], who were at significantly higher risk of hypoxemia or adverse events during bronchoscopy due to a higher incidence of hypoxemic events ($SpO_2 < 90\%$) than the other four trials (Fig 2), indicating the value of HFNC during invasive bronchoscopy in more vulnerable patients. Hence, a comprehensive algorithm for choosing HFNC or conventional oxygen therapy in patients undergoing bronchoscopy should be established, especially in patients with underlying severe lung disease. In the past, several studies have suggested that patients with (1) $SpO_2$ on room air pre-procedure $< 90\%$ [28]; (2) moderate sedation [29]; (3) obstructive sleep apnea (OSA) [30]; or (4) requiring long-term oxygen therapy, such as for congestive heart failure (CHF) [31], chronic obstructive pulmonary disease (COPD) [32], or interstitial lung disease (ILD) [33], may benefit from HFNC treatment. However, this need to be validated by more evidence.

Although there is substantial effort to detect the hypoxemic events in patients undergoing bronchoscopy, $PaCO_2$ or $EtCO_2$ are also important for clinicians to assess the lung condition. One previous study indicated the effectiveness of HFNC for $CO_2$ removal [27]; however, our systematic review showed that HFNC was not as effective as expected. All the trials presented lower $PaCO_2$ or $EtCO_2$ in the control group than in the HFNC group, except for the trial by Douglas et al. [14]. There are several reasons for this discrepancy. First, the number of trials was small. Second, these trials enrolled patients without severe pulmonary illnesses. A conclusive result cannot be obtained because the severity of pulmonary disease may influence the extent of $CO_2$ clearance. Hence, more evidence is required to evaluate the effectiveness of HFNC for $CO_2$ removal.

Several clinical factors across the trials induced heterogeneity in our meta-analysis. First, the approaches to intervention varied in terms of flow intensity and $FiO_2$. Second, different devices were used for the control groups, making exact $FiO_2$ measurement difficult, because it was influenced by the peak respiratory flow and the design of each low-flow system. Third, the

study population in the trial by Ben-Menachem et al. differed from that of the other trials [13]. Fourth, the use of sedative or anesthetic agents varied among the trials. To take such diversity into account, we utilized sensitivity analysis, with results shown in the S4 Appendix.

Our study has several limitations. First, the trials had small sample sizes for each group. Hence, the certainty of imprecision was downgraded accordingly. Second, the number of RCTs in patients with HFNC undergoing bronchoscopy was insufficient for conducting comprehensive analyses. Third, there is still debate over the routine use of oxygen supplementation during bronchoscopy [28]. Not all centers gave oxygen to patients, unless desaturation occurs. Fourth, we did not obtain $PaCO_2$ and $EtCO_2$ values during bronchoscopy, which prevented us from performing a more precise analysis. Fifth, according to the inclusion criteria, no suspected cancer patients or patients at high specified pulmonary risk were included in our systematic review, limiting the external validity of the results.

## Conclusion

Our meta-analysis revealed that HFNC may provide consistent oxygenation ($SpO_2 \geq 90\%$) and safer invasive procedures in patients undergoing bronchoscopy when compared with conventional oxygen therapy.

## Supporting information

**S1 Appendix. Search strategy.**
(DOCX)

**S2 Appendix. PRISMA checklist.**
(DOCX)

**S3 Appendix. Trial sequential analysis.**
(TIF)

**S4 Appendix. Sensitivity analysis.**
(DOCX)

## Acknowledgments

We acknowledge Editage Academic Editing for editing this manuscript.

## Author Contributions

**Conceptualization:** Chien-Ling Su.

**Data curation:** Tzu-Tao Chen.

**Investigation:** Ka-Wai Tam, Tzu-Tao Chen.

**Methodology:** Ka-Wai Tam.

**Supervision:** Chien-Ling Su.

**Writing – original draft:** Ming-Chi Hu.

**Writing – review & editing:** Ling-Ling Chiang, Ka-Wai Tam.

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
