## [Decision Letter · Decision Letter 0]

11 Aug 2021

PONE-D-21-21235

High-flow nasal cannula for reducing hypoxic events in patients receiving bronchoscopy: A systematic review and meta-analysis of randomized trials

PLOS ONE

Dear Dr. Hu,

Thank you for submitting your manuscript to PLOS ONE. After careful consideration, we feel that it has merit but does not fully meet PLOS ONE’s publication criteria as it currently stands. Therefore, we invite you to submit a revised version of the manuscript that addresses the points raised during the review process.

We look forward to receiving your revised manuscript.

Kind regards,

Alessandro Putzu, M.D.

Academic Editor

PLOS ONE

Additional Editor Comments :

A number of issues have been identified in the review process. While we feel that this manuscript shows promise, we also think that a major revision is needed. Before we can make a final decision about this manuscript we want to offer you the opportunity to revise and resubmit the manuscript.

1- Methods. Please include the new PRISMA 2020 checklist in the supplementary material.

2- Methods. The 4 search strategies should be reported in the supplementary material.

3- Methods. How did you manage missing outcome data? Did you contact corresponding authors?

4- Methods. You should consider to perform a trial sequential analysis for the primary outcome to assess risk of type 1 and 2 errors.

5- Results. The list of 27 major exclusions should be reported in the supplement (reference + reason for exclusion).

6- Results. Instead of Figure 1, please use a PRISMA 2020 flow-chart.

7- Results. Systematic searches found only 157 records. This is correct?

8- Results. You reported that all trials had low risk in the reporting bias item. Do you confirm that all trials had a registered protocol?

9- Results. You reported that all trials had low risk in performance bias. Do you confirm that all studies had ITT analysis?

10- Results. Methodology on GRADE should be moved to the Methods.

11- Results. The GRADE assessment for the primary outcome and lowest SpO2 had no serious concern on imprecision. However, only 4 RCTs reported the outcome.

12- Results. The studies were performed under local anesthesia? Sedation?

13- Discussion. The section should be tempered; GRADE results should be considered.

14- Discussion. The inclusion/exclusion criteria of the RCTs limit the external validity of the results. Please discuss.

15- Abstract. Conclusions of the abstract should be tempered since the certainty of evidence is moderate/low.

Reviewers' comments:

Reviewer's Responses to Questions

**Comments to the Author**

1. Is the manuscript technically sound, and do the data support the conclusions?

Reviewer #1: Yes

Reviewer #2: Yes

Reviewer #3: Partly

Reviewer #4: Yes

Reviewer #5: Partly

2. Has the statistical analysis been performed appropriately and rigorously? 

Reviewer #1: Yes

Reviewer #2: Yes

Reviewer #3: Yes

Reviewer #4: Yes

Reviewer #5: N/A

3. Have the authors made all data underlying the findings in their manuscript fully available?

Reviewer #1: Yes

Reviewer #2: Yes

Reviewer #3: Yes

Reviewer #4: Yes

Reviewer #5: Yes

4. Is the manuscript presented in an intelligible fashion and written in standard English?

Reviewer #1: No

Reviewer #2: Yes

Reviewer #3: Yes

Reviewer #4: Yes

Reviewer #5: Yes

5. Review Comments to the Author

Reviewer #1: I have reviewed the manuscript that is interesting. However, there are some limitation or concerns requiring discussion.

Comments:

1. English requires an extensive revision by a mother language author or service.

2. Intro: “Sedation and the occlusion of the bronchi during the procedure reduce the respiratory drive and alter the lung condition.” The reviewer agree that sedation may alter the respiratory drive or timing (refer to PMID: 23982026 and 28673877); however, it remains unclear what authors mean for “lung condition”. Please specify.

3. Intro: this section requires more quotation to support all statements.

4. Aim of the study: “we compared the efficacy and safety of HFNC with conventional oxygen”. Please specify what is efficacy and safety, even in the material and methods section

5. I would suggest to add a “take-home message” regarding possible indication on the use of HFNC over COT during bronchoscopy.

Reviewer #2: The study is useful for end-users and the process is well-conducted. The PICO selection should appear in the method section.

Introduction: well balanced

Method: appropriate

Results: standard

Discussion: moderately interesting

Conclusion: based on results

Illustrations: standard

Reviewer #3: This is a registered systematic review & meta-analysis of five RCTs that aimed to evaluate whether high flow therapy (HFT) is more effective at preventing hypoxic events during bronchoscopy. It is a well written manuscript that addresses an important clinical area, however a number of issues must be addressed prior to publication.

Major:

- A definition of hypoxic events must be provided in the methods and referred to throughout the results and discussion, with a clear justification for the chosen definition. Without this, an ambiguous reference to hypoxic events does not have clinical relevance and cannot contribute meaningfully to the evidence base.

- Descriptions of physiological effects of HFNC are clear and useful, however no references have been provided for each effect. Please refer to the relevant clinical and bench studies for each point made (consider reviewing PMID: 33664838 for guidance).

- Ambiguous references to potential adverse events during/following bronchoscopy (in the introduction and discussion) should be replaced with specific outcomes and references.

Minor:

Grammatical and style suggestions have been provided below, which would aid clarity.

ABSTRACT

- Line 26 Suggest “undergoing” rather than “receiving” bronchoscopy

- Use comma not semi-colon when listing results

INTRODUCTION

- First two sentences too ambiguous, need to provide more detail on potential complications of bronchoscopy (remove “respiratory disturbance” and “alter the lung condition”).

- There is not “usually” a high risk of hypoxic events requiring intervention – this is rare during elective procedures.

- What evidence is there that hypoxaemia and respiratory distress are debilitating? Please provide references.

- Line 69 – cardiothoracic surgeons may also perform bronchoscopy, remove “physicians”

- “Although oxygen therapy…” is a nebulous statement, remove or make more specific with dates and a reference.

- Conventional oxygen does not have limited effectiveness (line 73), the point here is that HFNC may be more effective.

- HFNC has many, many more applications than treating hypoxaemic respiratory failure (lines 77-78). Either remove statement or provide specific applications (summarised in aforementioned review PMID: 33664838)

- Do not reference one article for its physiological effects, individual trials need to be referenced for each point.

- Line 82 – not strictly true, the Service paper referenced, which is a feasibility study, did report conclusive results.

- Rephrase line 85 “Thus, through a SR…” to “The aim of this systematic review was to…” and list aims.

METHODS

- Inclusion criterion 3 is unclear – what is meant by “clearly” reporting eligibility criteria? Or medical treatments regimens or severity among the population? Needs to be more specific.

- Hypoxic events must be given a clear definition, particularly since this is the primary aim of the systematic review.

- Outcomes – use commas, not semicolons

RESULTS

- What are the irrelevant trials? This needs to be more specific.

- Line 169 “recruited patients of middle-to-old age” is not appropriate. Either provide the mean ± SD / median (IQR) age or omit.

- Line 169 – what is the relevance of including how studies reported baseline oxygenation? In the results, you describe that 4 studies use SpO2.

- Reference 12 is a very specific population at significantly higher risk of hypoxaemia/adverse events – this needs to be commented on.

- The allocation bias to malignancy is surely acceptable given this procedure is commonly performed to investigate suspected cancer?

- Risk of bias needs to be reported as low, high or unclear, not “Some concerns” (see Cochrane guidance), or the “some” needs to be elaborated on.

- Line 190 – hypoxic events has not been defined. This must be described in the methods for these analyses to have any relevance.

- Line 213 and 223 - “mainly” is inappropriate, an exact description of what was analysed should be reported.

- Incidence of intubation needs to be reported even if not statistically significant.

DISCUSSION

- Line 259 – “lower” not “low” incidence.

- It is crucial that a definition of hypoxaemic event be defined. Without this, the discussion and indeed the study cannot provide a meaningful contribution.

- Line 267 – “respiratory deficits” is unusual wording, consider rephrasing and being more specific.

- Line 270 – what evidence is there that HFNC improves “bronchial hygiene” and what does this mean? Whole paragraph requires references for each physiological effect.

- Line 276 – “hazardous hypoxic events” – definition needed.

- Line 294 – this is not correct. Based on your analyses, it cannot be concluded that HFNC reduces CO2 clearance.

- Paragraph starting line 300 – additional limitation by population of lung transplant data which likely heavily influence results given baseline morbidity.

- Limitations – elaboration needed on the small sample size and how this affects interpretation of results and clinical application. Define “acceptable oxygenation” and why this is an important point in the context of this review.

CONCLUSION

- This conclusion cannot be drawn based on the presented data.

- Needs to be much more comprehensive, cover the primary and secondary objectives, summarise limitations and provide suggestions to improve the quality of current evidence base.

Acknowledgement – what editing was done?

Reviewer #4: Dear authors,

This is an interesting article and addresses an important issue (hypoxemia during bronchoscopy). I have following comments for the manuscript.

1. The statement “Clearly reported patient inclusion and exclusion criteria, medical treatment regimens, severity among the population, and the definition and evaluation of hypoxic events during or after bronchoscopy were included in the analysis.” is not clear. Please rephrase it to clarify.

2. In CONSORT diagram – Records screen 136 and irrelevant trial 105, the number should be 31, instead of 32.

3. Please maintain uniformity in the figures – favour HFNC and favour COT should be consistently on same side. In figure 2 favour HNFC in right side whereas in figure 3 it is on other side.

4. English needs editing

5. Authors should also highlight that there is still debate over routine use of oxygen supplementation during bronchoscopy. Even at some centres, oxygen is only given to patients who have oxygen saturation of less than 96%.

6. Authors, did not comment whether is any risk group which would get more benefits with HFNC.

Reviewer #5: Comments to Author:

I think this study is highly novel regarding whether conventional oxygen therapy (COT) or high flow nasal canula (HFNC) is more effective in preventing hypoxemia for patients with during receiving bronchoscopy. However, there are several comments for this study. I am sorry, but I haven't read any further than the results, as I think there are serious problems with the methodology.

Major points

1. What is your research question? Is the presence or absence of sedation during bronchoscopy part of your question? I think the research question needs to be reconsidered because the presence of sedation is considered very important in the development of hypoxemia. In addition, the presence or absence of sedation can be a source of very large conceptual heterogeneity. Therefore, as I will point out below, I think it is highly likely that the search formula, inclusion and exclusion criteria will need to be reconsidered.

2. Line 62-: How often is sedation used during bronchoscopy? If you want to discuss about hypoxemia during bronchoscopy associated with sedation, I think the frequency of sedation needs to be described.

3. Line 67-: What exactly does “debilitating” mean? Have adverse events associated with hypoxemia been discussed in previous studies? If you want to express the need for oxygen administration, I think it would be more persuasive if you describe a specific description of the serious adverse events associated with hypoxemia.

4. Line 77-: In this paragraph, I think it is necessary to subscribe, for example, that recommendations may change because multiple randomized controlled trials (RCTs) have been reported but no systematic review (SR) or meta-analysis (MA) have not been integrated done. It seems to me that you are now describing why we must do RCTs.

5. Line 95-: If your research question includes the presence or absence of sedation during bronchoscopy, then I think you need to add the presence or absence of sedation to your inclusion or exclusion criteria.

6. Line 101-: Have you done any searches about ongoing studies? In the methods section, the authors stated that ClinicalTrials.gov registry was searched for ongoing trials. However, there was no results of the search in result section. Please explain this point.

7. Line 103-: Is it correct to assume that the following search formula was used: “HFNC” OR “high flow nasal cannula” OR “high flow nasal CPAP” OR “high flow nasal oxygen” AND “bronchoscopy” OR “bronchoscope”. Is this search term sufficient for the search formula? First, it is true that there is no MESH term of HFNC, but bronchoscopy of MESH term. When you search with your search formula, you cannot find all the synonyms, plurals, etc. contained in the MESH term. Second, If the search is to be done as you intend, then I think the following use of () is correct: (“HFNC” OR “high flow nasal cannula” OR “high flow nasal CPAP” OR “high flow nasal oxygen”) AND (“bronchoscopy” OR “bronchoscope”). Third, if you are targeting an RCT, you should use the study design filters which Cochrane recommended. I think that your current search formula is inadequate and does not allow you to perform the correct search. If you can't do the correct search, SA and MA will be completely useless.

8. Line 111-: I think it is necessary to extract and consider the procedure time for bronchoscopy. There is a significant correlation between the procedure time for bronchoscopy and the presence of hypoxemia. If it is not mentioned in each study, then I think it should be mentioned as such in your study.

9. Line 125-: What exactly is the definition of “the incidence of hypoxic events during bronchoscopy” in your study? Is it correct that you consider the outcome of each of the RCTs included in this MA as "the incidence of hypoxic events during bronchoscopy"? I think PaCO2 at the

end of bronchoscopy and EtCO2 at the end of bronchoscopy are different from “the incidence of hypoxic events during bronchoscopy”. After establishing specific clinically meaningful outcomes for hypoxemia, you should decide whether you can integrate the results of each study.

6. PLOS authors have the option to publish the peer review history of their article (what does this mean?). If published, this will include your full peer review and any attached files.

Reviewer #1: No

Reviewer #2: **Yes: **Marc Leone

Reviewer #3: No

Reviewer #4: No

Reviewer #5: No

---

## [Author Response · Author response to Decision Letter 0]

9 Sep 2021

Dear Editors and reviewers :

We wish to submit the revised version of our manuscript “High-flow nasal cannula for reducing hypoxemic events in patients undergoing bronchoscopy: A systematic review and meta-analysis of randomized trials” (PONE-D-21-21235) for publication in PLoS One. The paper was coauthored by Chien-Ling Su, Ling-Ling Chiang, and Ka-Wai Tam. 

We sincerely appreciate your review of our manuscript, and we hope that our edits and the responses we provide below satisfactorily address all the issues and concerns you and the reviewers have noted, and that the revised manuscript will now be suitable for publication in your journal.

As stated in the original submission, this manuscript has not been published or presented elsewhere in part or in entirety and is not under consideration by another journal. We have read and understood your journal’s policies, and we believe that neither the manuscript nor the study violates any of these. There are no conflicts of interest to declare.

Thank you for your consideration. We look forward to hearing from you.

Sincerely,

Tzu-Tao Chen and Ming-Chi Hu

Department of Pulmonary Medicine, Shuang Ho Hospital, Taipei Medical University

291, Zhongzheng Road, Zhonghe District, New Taipei City, 23561, Taiwan

Tel: 886-2-22490088 ext. 1251

E-mail: 09330@s.tmu.edu.tw (Tzu-Tao Chen)

E-mail: 20549@s.tmu.edu.tw (Ming-Chi Hu)

---

## [Decision Letter · Decision Letter 1]

26 Oct 2021

PONE-D-21-21235R1High-flow nasal cannula for reducing hypoxemic events in patients undergoing bronchoscopy: A systematic review and meta-analysis of randomized trialsPLOS ONE

Dear Dr. Hu,

Thank you for submitting your manuscript to PLOS ONE. After careful consideration, we feel that it has merit but does not fully meet PLOS ONE’s publication criteria as it currently stands. Therefore, we invite you to submit a revised version of the manuscript that addresses the points raised during the review process.

We look forward to receiving your revised manuscript.

Kind regards,

Andrea Cortegiani, M.D.

Academic Editor

PLOS ONE

Journal Requirements:

Additional Editor Comments (if provided):

Your modifications improved the manuscript. However, please address the comments from the comments from the reviewers which I agree.

Reviewers' comments:

Reviewer's Responses to Questions

**Comments to the Author**

1. If the authors have adequately addressed your comments raised in a previous round of review and you feel that this manuscript is now acceptable for publication, you may indicate that here to bypass the “Comments to the Author” section, enter your conflict of interest statement in the “Confidential to Editor” section, and submit your "Accept" recommendation.

Reviewer #1: All comments have been addressed

Reviewer #2: All comments have been addressed

Reviewer #3: All comments have been addressed

Reviewer #5: All comments have been addressed

2. Is the manuscript technically sound, and do the data support the conclusions?

Reviewer #1: Yes

Reviewer #2: Yes

Reviewer #3: Yes

Reviewer #5: Partly

3. Has the statistical analysis been performed appropriately and rigorously? 

Reviewer #1: Yes

Reviewer #2: Yes

Reviewer #3: Yes

Reviewer #5: Yes

4. Have the authors made all data underlying the findings in their manuscript fully available?

Reviewer #1: Yes

Reviewer #2: Yes

Reviewer #3: Yes

Reviewer #5: No

5. Is the manuscript presented in an intelligible fashion and written in standard English?

Reviewer #1: Yes

Reviewer #2: Yes

Reviewer #3: Yes

Reviewer #5: Yes

6. Review Comments to the Author

Reviewer #1: I have appreciated the effort of authors to improve the manuscript. I have further comments to be addressed

Reviewer #2: The authors responded to my comments and the study can help the readers. The manuscript is sound and seems ready for publication.

Reviewer #3: Many thanks considering the recommended revisions, which have all been included to provide further clarity and scientific rigor. This is an interesting and useful manuscript.

Reviewer #5: Comments to Author:

Thank you for your sincere response to my comment. I think your revision has made the manuscript of higher quality. However, there are still a few things that need to be revised.

Major points

1. Line 90-: After reading your response, I understand your opinion about not changing the search formula regarding sedation. However, I think it is necessary to include the presence or absence of general anesthesia, including midazolam and propofol, etc. , in the inclusion or exclusion criteria. In other words, the integration of all results should be limited to either local or general anesthesia, or integrated in each. As I pointed out in a previous review, the methods of sedation can be a source of very large conceptual heterogeneity.

2. the section of Results: I think it would be easier to understand if the results of each studies were summarized in a table, which could be added to table 1.

3. the section of Results: In the study of Ben-Menachem [13], the procedure time for bronchoscopy is about 30 minutes, which is about twice as long as the procedure time of other studies. Therefore I considered the possibility that there might be the effect of the study of Ben-Menachem [13] to the results, but since you have done a sensitivity analysis, I am satisfied regarding the results of your analysis of outcomes.

4. Line 194-: You responded, "we considered that PaCO2 and EtCO2 were important for respiratory system assessment". As you said, I also think that EtCO2 and PaCO2 are important for respiratory system assessment. However, I don't think high EtCO2 and SpO2<90% are synonymous. In fact, in the table 1 of the study of Lucangelo [17], the PaO2 for all groups was not less than 60 mmHg and therefore you cannot determine the SpO2 was less than 90%. Therefore, I think the study of Lucangelo [17] should be excluded from the analysis of the primary outcome.

7. PLOS authors have the option to publish the peer review history of their article (what does this mean?). If published, this will include your full peer review and any attached files.

Reviewer #1: No

Reviewer #2: No

Reviewer #3: No

Reviewer #5: No

---

## [Author Response · Author response to Decision Letter 1]

12 Nov 2021

Dear Reviewers:

We sincerely appreciate your further review of our manuscript (Ref: PONE-D-21-21235), entitled "High-flow nasal cannula for reducing hypoxemic events in patients undergoing bronchoscopy: A systematic review and meta-analysis of randomized trials." 

We have carefully addressed all the reviewers’ comments in our revised manuscript. The main corrections and point-by-point responses to the reviewer comments are provided below, with all corresponding changes marked in red font in the manuscript. 

We hope that our responses and revisions have adequately addressed the reviewers’ concerns, and that the revised manuscript will now meet the high standards required for publication in your esteemed journal. We look forward to hearing from you.

Sincerely yours,

Tzu-Tao Chen and Ming-Chi Hu

Department of Pulmonary Medicine, Shuang Ho Hospital, Taipei Medical University

291, Zhongzheng Road, Zhonghe District, New Taipei City, 23561, Taiwan

Tel: 886-2-22490088 ext. 1251

E-mail: 09330@s.tmu.edu.tw (Tzu-Tao Chen)

E-mail: 20549@s.tmu.edu.tw (Ming-Chi Hu)

---

## [Editor Report · Decision Letter 2]

16 Nov 2021

High-flow nasal cannula for reducing hypoxemic events in patients undergoing bronchoscopy: A systematic review and meta-analysis of randomized trials

PONE-D-21-21235R2

Dear Dr. Hu,

We’re pleased to inform you that your manuscript has been judged scientifically suitable for publication and will be formally accepted for publication once it meets all outstanding technical requirements.

Kind regards,

Andrea Cortegiani, M.D.

Academic Editor

PLOS ONE
---

## [Editor Report · Acceptance letter]

18 Nov 2021

PONE-D-21-21235R2 

High-flow nasal cannula for reducing hypoxemic events in patients undergoing bronchoscopy: A systematic review and meta-analysis of randomized trials 

Dear Dr. Hu:

I'm pleased to inform you that your manuscript has been deemed suitable for publication in PLOS ONE. Congratulations! Your manuscript is now with our production department. 

Kind regards, 

on behalf of

Dr. Andrea Cortegiani 

Academic Editor

PLOS ONE